# Proteomic profiling of urinary extracellular vesicles differentiates breast cancer patients from healthy women

Nilobon Jeanmard[1], Rassanee Bissanum[1], Hutcha Sriplung[2], Sawanya Charoenlappanit[3], Sittiruk Roytrakul[3]*, Raphatphorn Navakanitworakul[1]*

**1** Department of Biomedical Sciences and Biomedical Engineering, Faculty of Medicine, Prince of Songkla University, Songkhla, Thailand, **2** Department of Epidemiology, Faculty of Medicine, Prince of Songkla University, Hat Yai, Thailand, **3** National Center for Genetic Engineering and Biotechnology (BIOTEC), National Science and Technology Development Agency (NSTDA), Pathum Thani, Thailand

* sittiruk@biotec.or.th (SR); nraphatp@medicine.psu.ac.th (RN)

## Abstract

Urinary extracellular vesicles (uEVs) reflect the biological conditions of the producing cells. The protein profiling of uEVs allow us to better understand cancer progression in several cancers such as bladder cancer, prostate cancer and kidney cancer but has not been reported in breast cancer. We have, herein, aimed at quantifying the concentration and at generating the proteomic profile of uEVs in patients with breast cancer (BC) as compared to that of healthy controls (CT). Urine samples were collected from 29 CT and 47 patients with BC. uEVs were isolated by using differential ultracentrifugation, and were then characterized by Western blotting and transmission electron microscopy. Moreover, a nanoparticle tracking analysis was used in order to measure the concentration and the size distribution of urine particles and uEVs. The proteomic profiling of the uEVs was facilitated through LC-MS/MS. The uEV concentration was not significantly different between the assessed groups. The undertaken proteomic analysis revealed 15,473 and 11,278 proteins in the BC patients' group and the CT group, respectively. Furthermore, a heat map analysis revealed a differential protein expression, while a principal component analysis highlighted two clusters. The volcano plot indicated 259 differentially expressed proteins (DEPs; 155 up- and 104 down-regulated proteins) in patients with BC compared with CT. The up-regulated proteins from BC-derived uEVs were enriched in pathways related to cancer progression (i.e., cell proliferation, cell survival, cell cycle, cell migration, carbohydrate metabolism, and angiogenesis). Moreover, we verified the expression of the upregulated DEPs using UAL-CAN for web-based validation. Remarkably, the results indicated that 6 of 155 up-regulated proteins (POSTN, ATAD2, BCAS4, GSK3β, HK1, and Ki-67) were overexpressed in BC compared with normal samples. Since these six proteins often act as markers of cell proliferation and progression, they may be potential biomarkers for BC screening and diagnosis. However, this requires validation in larger cohorts.

**Data Availability Statement:** The MS/MS raw data and analysis files have been deposited in the

ProteomeXchange Consortium (http://proteomecentral.proteomexchange.org) via the jPOST partner repository (https://jpostdb.org) with the data set identifier JPST001972 and PXD039122.

**Funding:** This research was supported by the National Science, Research and Innovation Fund (NSRF) and Prince of Songkla University (Grant No. SCI4693040) to RN. The funders had no role in study design, data collection and analysis, decision to publish, or preparation of the manuscript.

**Competing interests:** The authors have declared that no competing interests exist

## Introduction

Breast cancer (BC) is the most frequently occurring cancer type among women. According to GLOBOCAN 2020, the number of new cancer cases in women worldwide has risen to 19.3 million. In 2020, there were 2,261,419 patients diagnosed with BC, and BC was responsible for 15.5% of deaths. In Thailand, approximately 22,158 women were diagnosed with BC in 2020, accounting for 22.8% of total new cancer cases [1]. The high incidence and mortality of BC mean that early screening is essential to reduce BC-associated deaths [2].

According to the National Comprehensive Cancer Network guidelines, BC screening tests (including mammography, ultrasound, and magnetic resonance imaging) are recommended for detecting BC. Although, mammography is the gold standard for the early detection of BC, there are several risks associated with it (i.e., overdiagnosis, overtreatment, the obtaining of false-positive test results, an exposure to radiation, and the obtaining of a false-positive biopsy) [3]. Immunohistochemistry is widely used for the evaluation of BC markers such as the estrogen receptor (ER), the progesterone receptor, the human epidermal growth factor receptor 2, and the proliferation marker protein Ki-67, thereby facilitating the diagnosis and molecular subtyping of BC and guiding a more personalized treatment. However, the tumor biopsy is very invasive and, in most cases, unnecessary (when tumors are benign). A tumor biopsy may also cause pathological changes in the tumor, tumor displacement, and cancer metastasis [4]. Moreover, the obtained small piece of tissue might not be able to reflect the tumor's heterogeneity [5]. Furthermore, immunohistochemistry has certain drawbacks: it lacks quantitative measurements, it is a largely subjective procedure, and it is time-consuming [6, 7]. Blood biopsy has emerged as an auxiliary method that allows us to assess specific molecular biomarkers such as circulating tumor cells, circulating tumor DNAs, circular RNAs, long noncoding RNAs, and extracellular vesicles (EVs), in BC patients for screening, diagnostic, monitoring, treatment outcome prediction purposes [4]. Although the blood biopsy assessing circulating EVs seems to be minimal, but it is still invasive.

EVs are membrane vesicles released from various types of cells. These vesicles play an important role in cell-to-cell communication by transferring signaling components (such as proteins, nucleic acids, lipids, and transcriptional factors) and influencing biological processes in the recipient cells. Cancer-derived EVs may contain specific molecules associated with cancer progression and invasion through the promotion of the intercellular transferring of cargoes within the tumor microenvironment. In addition, EV cargoes might reflect the altered state of the original tumors [8], while the EV concentrations have been related to BC progression [9]. Moreover, previous studies have shown that EVs secreted from cancer cells can migrate to distant organs in order to form niches, thereby leading to the formation of metastases.

Recently, urinary EVs (uEVs) have emerged as potential non-invasive biomarkers of many urological cancers (i.e., kidney, bladder, and prostate cancer) [5], non-urological cancers (such as lung cancer) [10], Parkinson disease [11, 12], and Alzheimer disease [13]. However, the proteomic profiling of urinary EVs in BC patients have not yet been prevalently reported. A few studies have focused on miRNA, as well as on the combined expression of miR-21 and matrix metalloproteinase 1 (MMP-1) in urinary exosomes, as BC biomarkers. However, miRNA-21 and MMP-1 are not specific molecules for BC [14]. Urinary miRNAs have been reported as potential non-invasive biomarkers in BC detection through the combined detection of miR-424, miR-423, miR-660, and let7-i [15]. Besides, significantly altered miRNAs (miR-21, miR-125b, miR-451, and miR-155) signify a potential role of urinary miRNAs as non-invasive biomarkers for the detection of BC [16]. Nevertheless, these molecules can also be markers for other cancer types.

We, herein, deliver for the first time a proteomic profiling of uEVs in BC patients, and assess whether this proteomic profiling could differentiate BC patients from healthy women.

## Materials and methods

This pilot study aimed at measuring the uEV concentration and analyzing the proteomic profile of uEVs in BC patients (in comparison to that of healthy women). The urine samples were obtained from project REC.59-233-18-1, and were contributed by Associate Professor Dr. Hutcha Sriplung (Department of Epidemiology, Faculty of Medicine, Prince of Songkla University). The workflow of this study was showed in Fig 1. The mid-stream urine samples were collected at random times and placed in sterile containers (cat#4CLM-4020-0204-13). The urine samples were collected from newly diagnosed patients with BC before surgery enrolled at the Songklanagarind Hospital (n = 47; aged 18 years or older). For the control (CT) group, urine samples were obtained from 29 healthy women without a history of cancer that were also age-matched with the BC patients (Table 1). All volunteers provided informed and written consent. Participants who were pregnant, could not speak Thai, or suffered from mental illnesses were excluded from this study. The study protocol was reviewed and approved by ethics committee of the Faculty of Medicine of Prince of Songkla University, Thailand (approval number: REC 64-511-4-2).

### Urine preparation

The urine preparation protocol was a modification of that of Gheinani *et al.* [17], and urine samples were collected from healthy women and patients with BC. The samples were centrifuged at 2,500 g, for 20 min, at 4˚C in order to remove cells, cell debris, and bacteria. The supernatants were then collected and stored at -80˚C until use (Fig 1).

### Total particles and uEV quantification by Nanoparticle Tracking Analysis (NTA)

The concentration and the size distribution of the total particles in the urine and the uEVs were measured by NanoSight NS300 (Malvern Panalytical Ltd, Malvern, UK). Each sample was diluted in sterile water with two independent dilutions in the recommended concentration of $1 \times 10^7 - 10^9$ particles/mL, which is approximately 20−100 particles/frame. Subsequently, the mixture was infused into the nanoparticle tracking analyzer with a syringe pump at speed 50. Each replicate was recorded in five 30-sec videos at a controlled temperature of 25˚C, a camera level of 14, and a detection threshold of 6. The obtained data were analyzed with the use of the NanoSight NTA 3.0 software.

### uEV isolation by differential ultracentrifugation

The uEV isolation protocol was adapted from the studies of Sequeiros *et al.* [18] and of Barros *et al.* [19]. The prepared urine samples (9 mL) were thawed and centrifuged at 2,500 g, for 15 min, at 4˚C. After discarding the pellets, the supernatants were collected and centrifuged at 20,000 g, for 20 min, at 4˚C in order to remove large particles. The obtained supernatant (SN1) was collected and kept on ice. For the depletion of the Tamm–Horsfall protein (THP), the 20,000-g pellets resuspended with 400 µL of isolation solution (containing 10 mM triethanolamine and 250 mM sucrose) were vortexed for 30 sec, added with 100 µL of 200 mg/mL dithiothreitol (DTT), and incubated at 37˚C, for 10 min. During the incubation, the mixture was vortexed every 2 min, and supplemented with 500 µL of isolation solution. Subsequently, the mixture was centrifuged at 20,000 g, for 20 min, at 4˚C. The newly obtained supernatant was

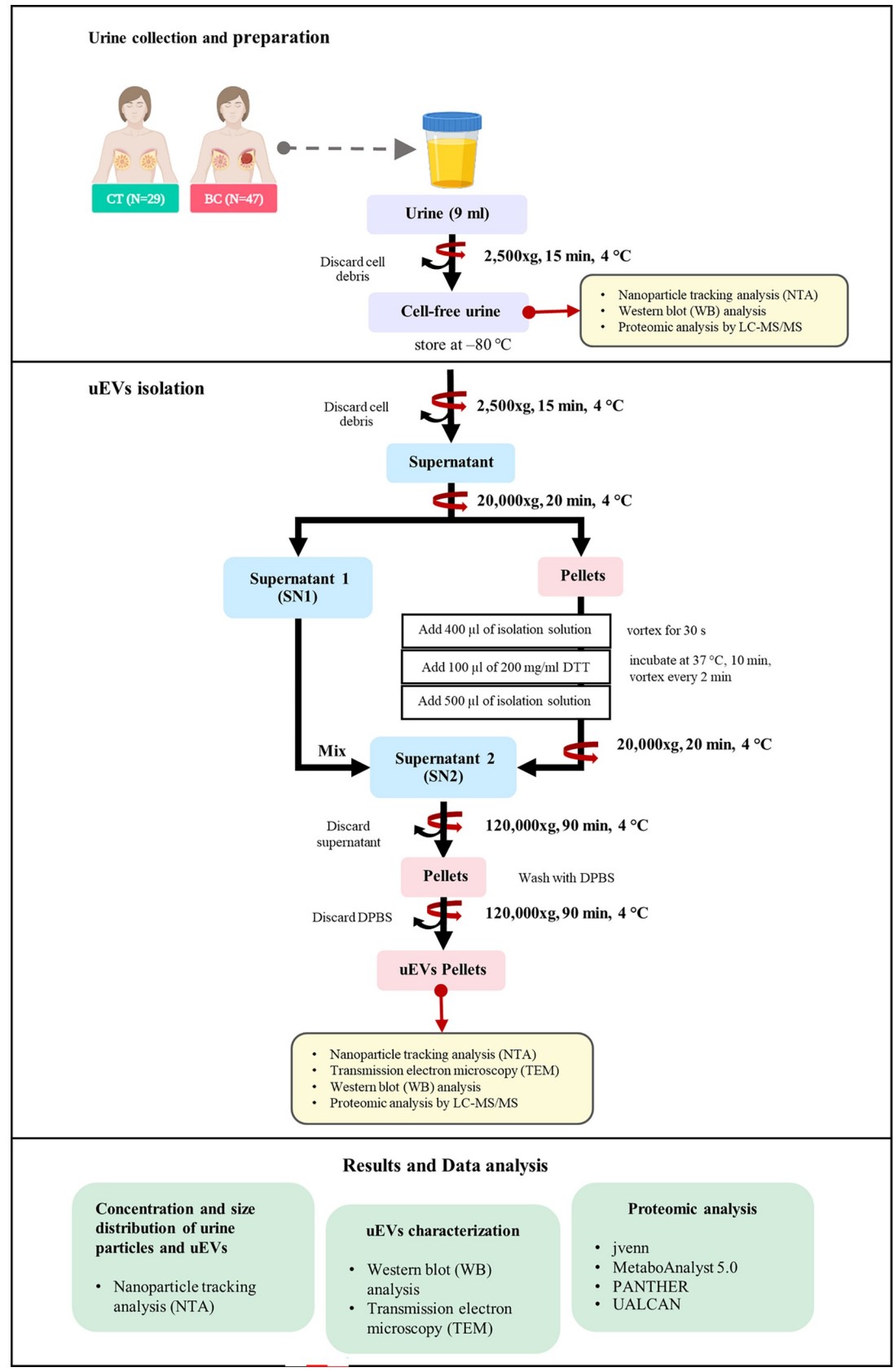

**Fig 1. Workflow for the investigation of the uEV proteome.**

**Table 1. Clinical characteristics of healthy controls and breast cancer patients.**

| Characteristics | Healthy controls (n = 29) | Breast cancer patients (n = 47) |
|---|---|---|
| **Age (Mean ± SD)** | 52.97 ± 3.76 | 53.49 ± 9.99 |
| <50 | 7 (24.14%) | 15 (31.91%) |
| ≥50 | 22 (75.86%) | 32 (68.09%) |
| **Subtypes** | | |
| Luminal A | - | 22 (46.81%) |
| Luminal B | - | 16 (34.04%) |
| HER2-enriched | - | 6 (12.77%) |
| TNBC | - | 3 (6.38%) |
| **Tumor size (cm)** | | |
| < 2 | - | 17 (36.17%) |
| 2–5 | - | 29 (61.70%) |
| > 5 | - | 1 (2.13%) |
| **Lymph nodes involvement** | | |
| 0 | - | 28 (59.57%) |
| 1–3 | - | 17 (36.17%) |
| 4–9 | - | 1 (2.13%) |
| ≥10 | - | 1 (2.13%) |
| **Stage** | | |
| I | - | 18 (38.30%) |
| II | - | 26 (55.32%) |
| III | - | 3 (6.38%) |
| **Side of BC** | | |
| Left breast | - | 24 (51.06%) |
| Right breast | - | 20 (42.55%) |
| Both sides | - | 3 (6.38%) |

collected and pooled with the SN1, before ultracentrifugation at 120,000 g, for 90 min, at 4°C (Beckman Coulter, Optima MAX-XP Ultracentrifuge). The pellets were washed with Dulbecco's PBS (DPBS) and ultracentrifuged at 120,000 g, for 90 min, at 4°C. Finally, the remaining pellets (uEV pellets) were resuspended in 150 µL of DPBS (Fig 1).

## Western blotting analysis

We mixed 10 µL of urine and uEV samples with non-reducing dye, boiled at 100°C for 5 min, and then kept at −20°C for 5 min. Urine and uEV proteins were separated through a SDS-PAGE gel by using the TGX Stain-Free™ FastCast™ Acrylamide Kit, 12% (cat#161–0185; Biorad) with running conditions of 80 V in stacking gel for 20 min, and 120 V in resolving gel for 1 h. Subsequently, the proteins were transferred to polyvinylidene difluoride membranes. The membranes were blocked with 5% non-fat milk in Tris-buffered saline containing 0.1% Tween 20 (TBS-T), for 1 h, at room temperature. The membranes were then washed three times with TBS-T, for 10 min and incubated overnight, at 4°C, with primary antibodies (1:1,000 dilution) against CD9, TSG101, THP, and cytochrome c. After the incubation, the membranes were washed three times with TBS-T, for 10 min, and further incubated for 2 h with horse radish peroxide-conjugated secondary antibodies (1:2,000) at room temperature. After washing the membranes, the protein bands were detected by using the SuperSignal™ West Dura Extended Duration Substrate (cat#34075; Thermo Fisher Scientific), and visualized through a chemiluminescence imager (Alliance Q9 Advanced, UVITEC).

### Transmission Electron Microscopy (TEM)

The morphology and the size of uEVs were characterized by TEM. The uEV pellets were fixed with 3 μL of 2.5% glutaraldehyde, for 30 min, at room temperature. Subsequently, 3 μL of each sample were dropped on carbon/formvar-coated grids and incubated for 10 min. The grids were then washed two times with PBS for 3 min, and ten times with distilled water for 2 min. Each uEV sample was negatively stained with 2.5% of uranyl-acetate for 10 min, and left to dry overnight at room temperature. The uEVs were visualized by a transmission electron microscope (JEOL JEM 2010) with 50,000× magnification at the Scientific Equipment Center of the Prince of Songkla University.

### Proteomic analysis

Total protein was isolated from individual urine and uEV samples and measured with a Lowry assay using bovine serum albumin as the standard [20]. The protein samples (5 μg) were reduced with 5 mM DTT at 60°C for 1 h and alkylated in 15 mM iodoacetamide at room temperature for 45 min in the dark. The protein samples were then digested with sequencing grade porcine trypsin (1:20 ratio) for 16 h at 37°C. The tryptic peptides were dried using a speed vacuum concentrator and resuspended in 0.1% formic acid for LC-MS/MS analysis.

The tryptic peptide samples were analyzed by LC-MS on an UltiMate 3000 Nano/Capillary LC System (Thermo Scientific) coupled to an Impact II™ hybrid quadrupole time-of-flight mass spectometer (Bruker Daltonics) equipped with a nano-captive spray ion source. The peptide digests were enriched using a μ-precolumn (C18 PepMap 100; Thermo Scientific) and separated on an Acclaim PepMap RSLC analytical column (C18, nanoViper; Thermo Scientific). The C18 column was enclosed in a thermostatted column oven set at 60°C. A gradient of 5%–55% solvent B (0.1% formic acid in 80% acetonitrile) was used to elute the peptides at a constant flow rate of 0.30 μl/min for 30 min. Electrospray ionization was carried out at 1.6 kV using the CaptiveSpray system (Bruker Daltonics). Nitrogen was used as a drying gas (flow rate: approximately 50 l/h). Collision-induced-dissociation product ion mass spectra were obtained using nitrogen gas as the collision gas. Mass spectra (MS) and MS/MS spectra were obtained in the positive-ion mode at 2 Hz over a range ($m/z$) of 50–2200. The collision energy was adjusted to 10 eV as a function of the $m/z$ value. LC-MS analysis of each sample was performed in triplicate.

MaxQuant v2.1.0.0 software was used to quantify the proteins in individual samples, and the Andromeda search engine was used to correlate MS/MS spectra with the UniProt *Homo sapiens* database [21]. Label-free quantitation was performed using the standard MaxQuant settings: mass tolerance of 0.6 Daltons for main search, trypsin as the digestive enzyme, a maximum of two missed cleavages, carbamidomethylation of cysteine residues as a fixed modification, and oxidation of methionine and acetylation of the protein N-terminus as variable modifications. Only peptides with a minimum of seven amino acids, as well as at least one unique peptide, were required for protein identification. The protein false discovery rate (FDR) was set at 1% and estimated using reversed search sequences. The maximal number of modifications per peptide was set at five.

The Venn diagram was constructed using jvenn [22] to assess the overlap of proteins between EV proteins identified in our study and three publicly available databases: EVpedia (https://evpedia.info/evpedia2_xe/), ExoCarta (http://www.exocarta.org/), and Vesiclepedia (http://www.microvesicles.org/). These databases were accessed on June 13, 2023. To visualize the protein expression, a volcano plot and a heat map were generated using MetaboAnalyst 5.0 [23]. Additionally, principal component analysis (PCA) was performed using the same software.

To classify the molecular functions, biological processes, and pathways of the identified proteins, we utilized the PANTHER database [24]. Furthermore, a pathway analysis of the unique proteins (presented in the supplementary data) was performed using WebGestalt (https://www.webgestalt.org). We employed the Molecular Complex Detection (MCODE) plugin within Cytoscape v3.10.0 to identify the top cluster of protein–protein interactions annotated using the STRINGv11.5 database (https://string-db.org/) and analyzed pathways using Metascape (https://metascape.org). The analysis of the expressed proteins was facilitated by the integrated cancer data analysis platform UALCAN, by using data obtained from the Clinical Proteomic Tumor Analysis Consortium (CPTAC).

## Statistical analysis

All data were analyzed by using the GraphPad Prism 9.0.0 software. The uEV concentration was tested for the normality of its distribution, and results are described as median ± SEM. The statistical analysis between the unpaired two groups was performed by using a Mann–Whitney test. Values of $p$ that were found to be <0.05 were considered as indicative of statistical significance. The DEPs of the CT and BC groups were determined using the criteria of an FDR-adjusted $p$-value < 0.05 and fold change (FC) ≥4.

## Results and discussion

### The concentration and size distribution of total particles in urine and uEV samples

NTA revealed that the concentration of total particles in the CT group was $7.83 \times 10^9$ particles/mL; significantly higher than the concentration of the total particles measured in the BC patients' group, which was $4.74 \times 10^9$ particles/mL (Fig 2A). The size distribution of total particles in the urine were in a range of 101–150 nm (Fig 2B). Additionally, the mode size of the total particles in urine of the CT group and the BC patients' group were 97.55 and 97.3 nm, respectively (Fig 2C). On the other hand, the uEV concentrations of the CT group and the BC patients' group were $1.47 \times 10^{10}$ and $1.8 \times 10^{10}$ particles/mL, respectively. The uEV concentrations were not significantly different between the two groups (Fig 2D). The uEV size distribution was in the range of 101–150 nm in both the CT group and the BC patients' group (Fig 2E). Furthermore, the mode size of the uEVs in the CT group and in the BC patients' group were not significantly different with sizes of 96.2 and 96 nm, respectively, which were individually closely in size in each group when compared to that of urine samples before the uEV isolation process (Fig 2F). Even in terms of subtypes and stages of BC, uEV concentrations did not differ significantly among sample groups (S1 and S2 Figs).

### Characterization of uEVs

Western blotting analysis revealed that EV protein markers, such as CD9 and TSG101, were expressed in urine samples and enriched in the isolated uEV samples. Interestingly, the THP levels were found to decrease in uEVs (when compared to those of urine samples). Moreover, cytochrome c, as a negative control, was not detected in any of our samples (Fig 3A). The size and the morphology of the uEV pellets were examined by TEM. The results showed that the uEV pellets were round in shape and had a diameter within 30 to 150 nm (Fig 3B).

### The comparison among the uEV proteomes

We compared the 17,692 uEV proteins identified in this study with the proteins published in the EVpedia, ExoCarta, and Vesiclepedia databases. Our analysis revealed that 23.067% of the

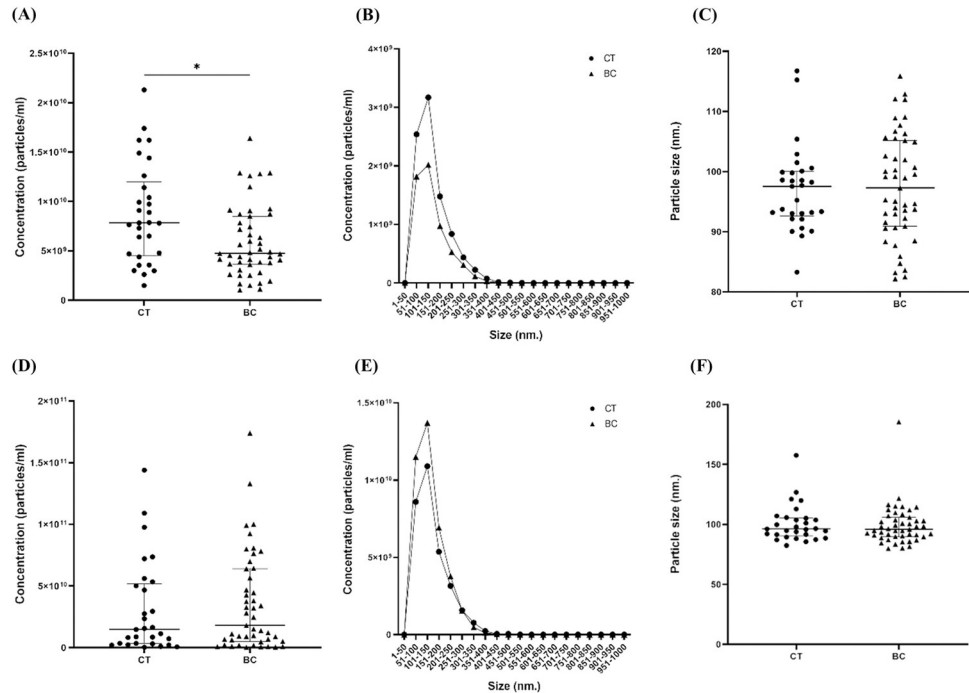

**Fig 2. The concentration of total particles in urine and uEV samples deriving from CT (n = 29) and BC patients (n = 47).** (A) Total particle concentration in urine (median ± IQR). (B) Total particle size distribution in urine. (C) Mode size of total particles in urine (median ± IQR). (D) uEV concentration (median ± IQR). (E) uEV size distribution. (F) Mode size of uEVs (median ± IQR). *: $p < 0.05$.

uEV proteins identified in our study were unique to our dataset, while 76.933% of the proteins were shared across all three databases (Fig 4). Additionally, the number of urine and uEV proteomes was compared between our dataset of patients with BC and healthy control samples. The total number of proteins in the uEVs was greater than that in urine. A total of 15,374 proteins were discovered in the uEVs of the patients with BC group, while 11,278 proteins were found in the uEVs of the CT group. However, 1,104 and 1,097 proteins were detected in the

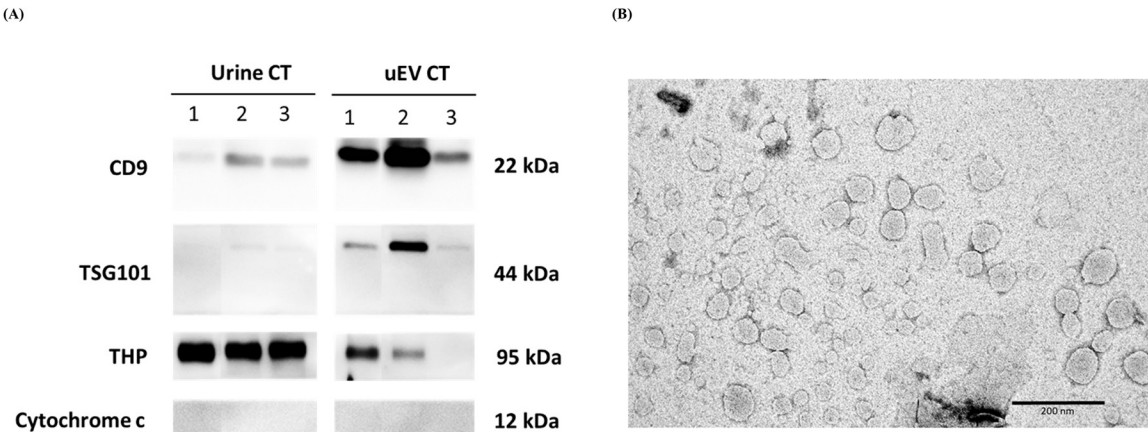

**Fig 3. Characterization of uEVs.** (A) Western blotting analysis of whole urine and uEV pellets obtained following ultracentrifugation at 120,000 g of control samples (n = 3). (B) TEM analysis of uEVs isolated by ultracentrifugation (magnification: 50,000×).

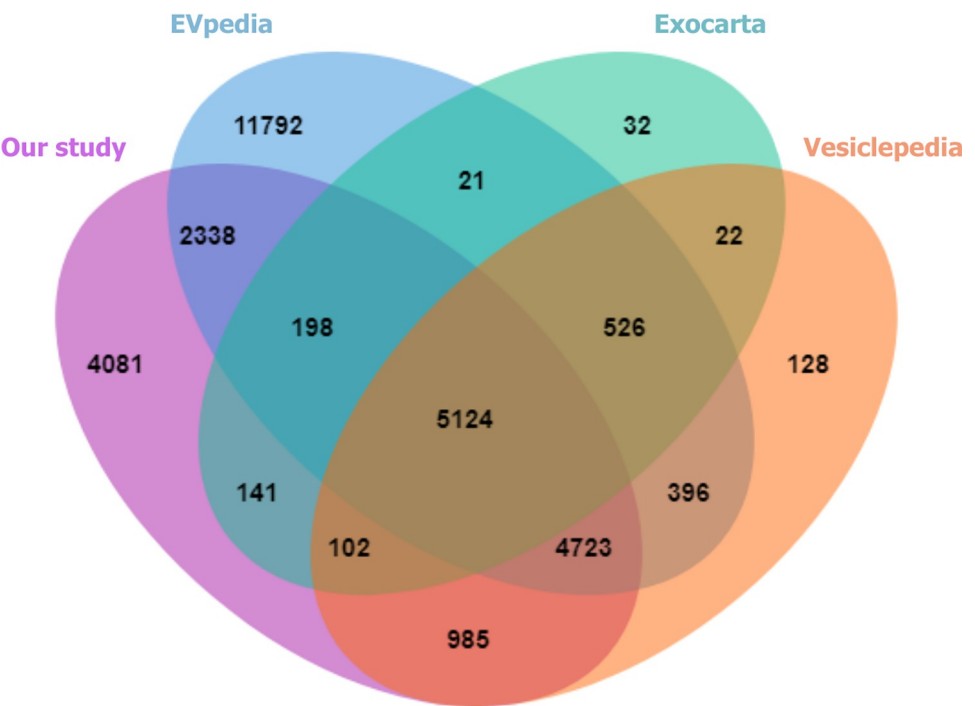

**Fig 4. Venn diagram showing all identified uEV proteins in this study compared with those published in the EVpedia, ExoCarta, and Vesiclepedia EV databases.**

urine of the BC and CT groups, respectively. Additionally, more than one thousand proteins in urine samples overlapped with the uEV proteins (Fig 5).

The uEV and urine proteomes were compared in terms of the number of proteins, the protein expression, and the clustering, as shown in Fig 6. The Venn diagram revealed the lower proportion of unique proteins in the urine samples (as compared to the unique proteins identified in the uEV samples). Moreover, the uEV proteins of the BC patients' group displayed 4,753 signature proteins, whereas the CT group exhibited 558 unique uEV proteins (Fig 6A and 6B). A heat map revealed the protein expression patterns in individual samples. The uEV proteome exhibited distinct protein expression patterns among BC patients and healthy women (CT) (Fig 6D). These patterns were more obvious than those observed in the urine proteome (Fig 6C). In addition, the PCA of individual samples categorized the uEV proteomic data in a 2D graph as PC1 (17.3%) and PC2 (6.6%), and identified two clusters that tended to be distinguishable between the BC and the CT groups (Fig 6F). In contrast, no separate clusters appeared after the application of a PCA on the urine proteomic data in PC1 (3.5%) and PC2 (2.7%) (Fig 6E).

### Differentially expressed proteins (DEPs)

In order to reveal the DEPs of the uEV samples, a volcano plot identified 259 significantly DEPs, of which 155 were up- and 104 were down-regulated proteins in BC patients as compared with healthy controls (Fig 7 and S1 Table). The fold change (FC) that was $\geq 4$ and an adjusted $p$-value that was $<0.05$ were set as the thresholds for the DEP determination.

### Functional analysis of DEPs

We then used the PANTHER database in order to demonstrate the molecular functions, the biological processes, and the pathways of 259 DEPs. The molecular functions of the up- and

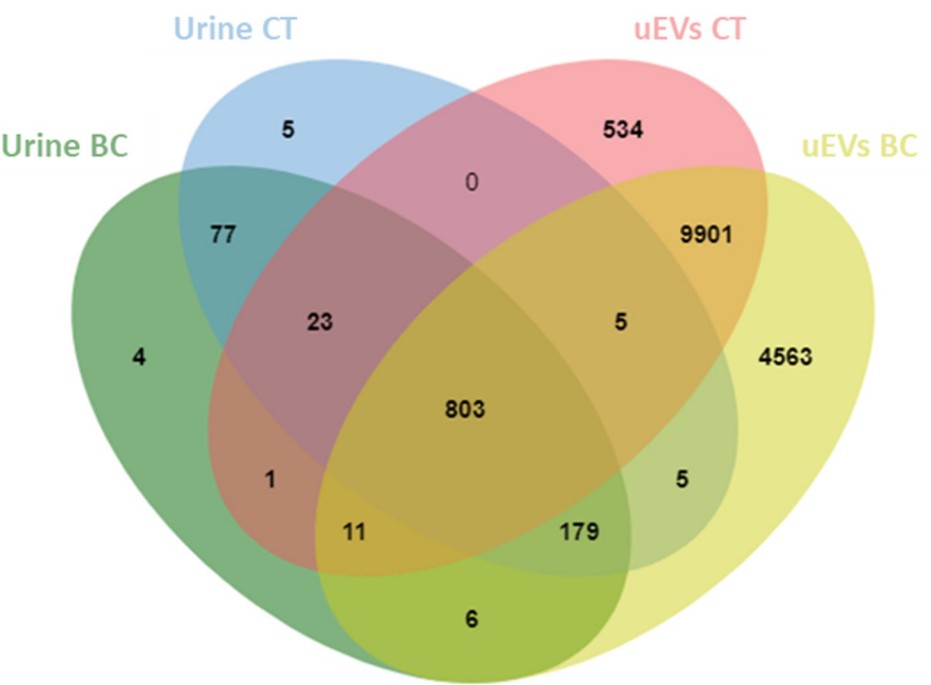

**Fig 5. Venn diagram of total proteins identified in the urine compared with uEVs in samples derived from BC patients and healthy controls (CT).**

the down-regulated DEPs were those of major proteins related to binding and catalytic activity, followed by transporting, transcription regulation, ATP-dependent activity, molecular function regulation, molecular transduction, and molecular adaptor activity. Besides, some up-regulated DEPs were associated with structural molecular activity and cytoskeletal motor activity (Fig 8A).

The biological processes of the up- and the down-regulated DEPs were performed generally by proteins involved in cellular, metabolic, and biological regulation. Subordinate categories were related to localization, response to stimulus, the developmental process, the signaling and the multicellular organismal process, biological adhesion, locomotion, and the immune system process. The remaining biological processes (such as the rhythmic process, the reproductive process, reproduction, and growth) were identified in association with the up-regulated DEPs, while the down-regulated DEPs were associated only with the interspecies interaction process between organisms (Fig 8B).

Furthermore, the undertaken pathway analysis has revealed up- and down-regulated proteins associated with a group of pathways primarily involved in cell proliferation, cell survival, cell cycle, and cell migration, the receptor-mediated signaling pathway, and the immune system. Interestingly, the pathways of the metabolism of carbohydrates and of angiogenesis were particularly reflected by the up-regulated proteins (Fig 8C). The group of pathways associated with cell proliferation, cell survival, and migration, along with that of the cell cycle, included many pathways such as the EGF receptor signaling pathway (P00018), the PI3 kinase pathway (P00048), the PDGF signaling pathway (P00047), the Ras pathway (P04393), the Notch signaling pathway (P00045), the CCKR signaling map (P06959), the hedgehog signaling pathway (P00025), the insulin/IGF pathway-protein kinase B signaling cascade (P00033), the integrin signaling pathway (P00034), the cadherin signaling pathway (P00012), the metabotropic glutamate receptor group I pathway (P00041), the metabotropic glutamate receptor group III

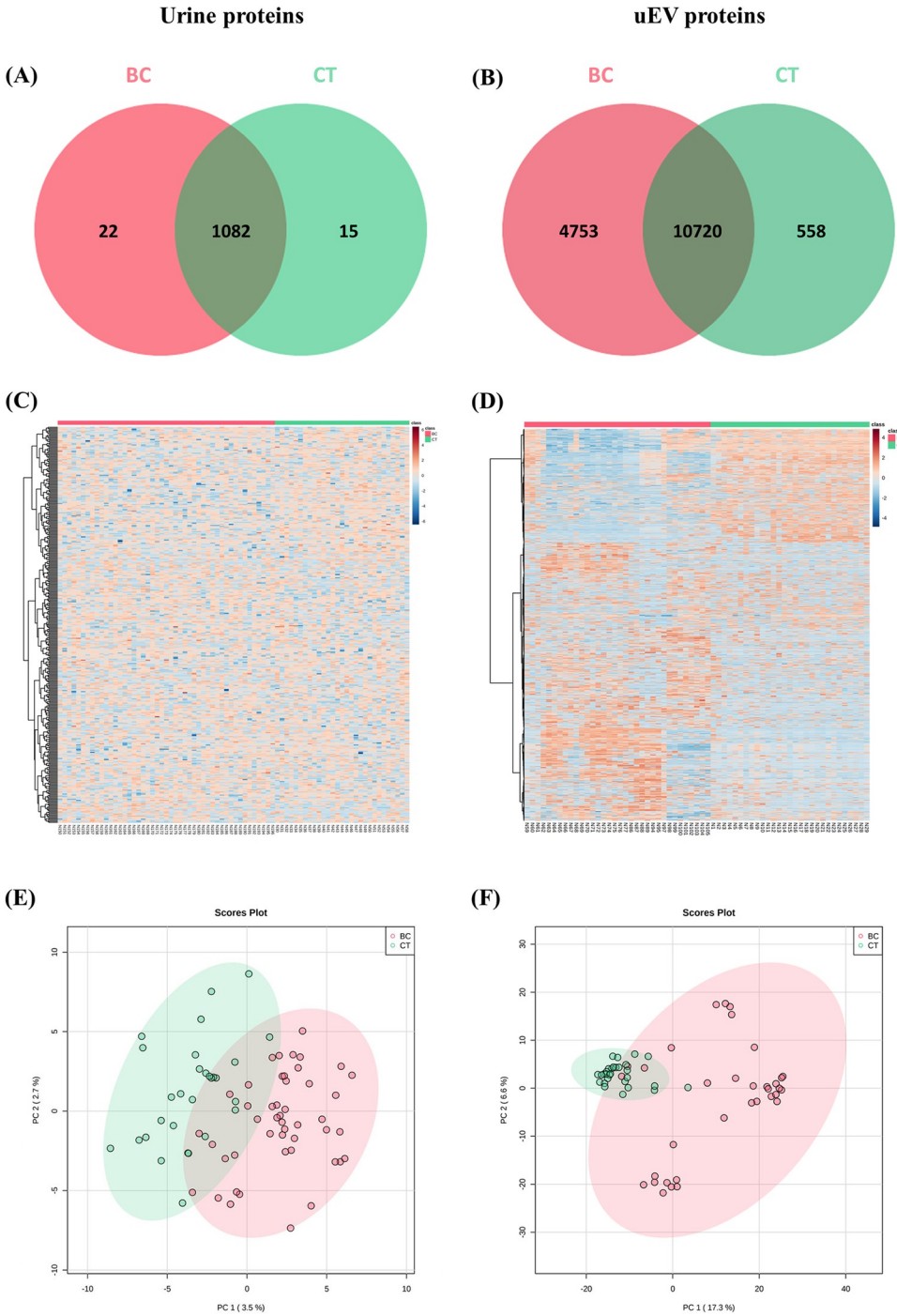

**Fig 6. The comparison of the urine and the uEV proteomic analysis.** (A) Venn diagram of the raw proteins identified in urine samples. (B) Venn diagram of the raw proteins identified in uEV samples. (C) Heat map analysis of the urine proteomic profile in individual samples. (D) Heat map analysis of the uEV proteomic profile in individual samples. (E) PCA of the proteome of urine obtained from patients with BC (n = 47) and CT (n = 29). (F) PCA of the proteome of uEVs obtained from patients with BC (n = 34) and CT (n = 29). The red color represents the patients with BC group, while the green color represents the CT group.

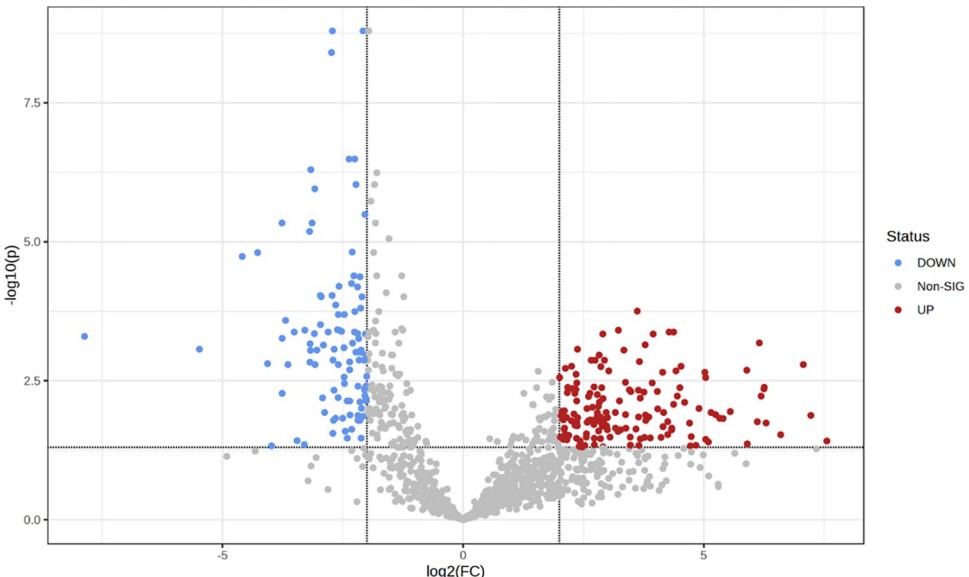

**Fig 7. Volcano plot showing the DEPs of the uEV proteome of BC patients compared with those of healthy females (CT).** The up-regulated proteins are shown as red dots, and the down-regulated proteins are shown as blue dots; their FC is ≥4, and their adjusted p-value is <0.05 (DEP thresholds).

pathway (P00039), the Wnt signaling pathway (P00057), the p53 pathway (P00059), the p53 pathway feedback loops 1 (P04392), the p53 pathway feedback loops 2 (P04398), and the gonadotropin-releasing hormone receptor pathway (P06664). In addition to the carbohydrate metabolism pathway, those of glycolysis (P00024), fructose galactose metabolism (P02744), and pentose phosphate pathway (P02762) were found only in association with the up-regulated DEPs (S2 and S3 Tables).

Consequently, a network was generated to explain the relationship between proteins and their pathways, particularly pathways of cancer progression, carbohydrate metabolism, and angiogenesis, as annotated by the PANTHER database (Fig 9).

## Web-based validation of up-regulated proteins using the UALCAN database

Expression of the up-regulated proteins derived from uEVs of patients with BC was further validated using the CPTAC dataset of BC in the UALCAN database. We found that 6 of 155 up-regulated DEPs were significantly overexpressed in primary breast tumors compared with normal samples, including periostin (POSTN), ATPase family AAA domain-containing protein 2 (ATAD2), breast carcinoma-amplified sequence 4 (BCAS4), glycogen synthase kinase 3 beta (GSK3β), and hexokinase 1 (HK1). Additionally, Ki-67, as a marker of cell proliferation, was expressed at significantly higher levels in triple-negative breast cancer (TNBC) than in other BC subtypes and normal samples (Fig 10 and S4 Table).

## Identification of subtype-specific uEV proteins in BC

The uEV proteins in the control group were compared with the four BC subtypes to identify uEV proteins specific to each subtype. The number of unique proteins (Fig 11A) was 558 for the CT group and 752, 154, 619, and 587 for the luminal A, luminal B, HER2, and TNBC subtypes, respectively (S5 Table). Heat map revealed distinct patterns of protein expression

**(A)**

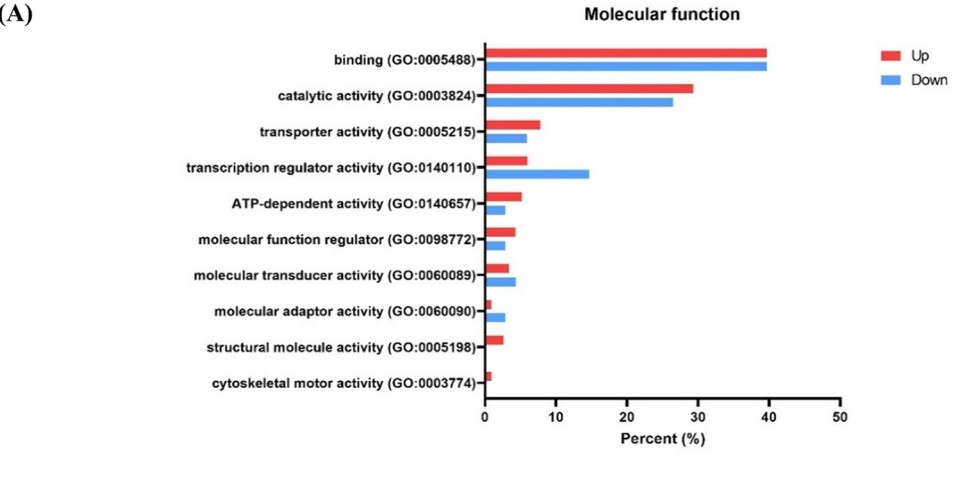

**(B)**

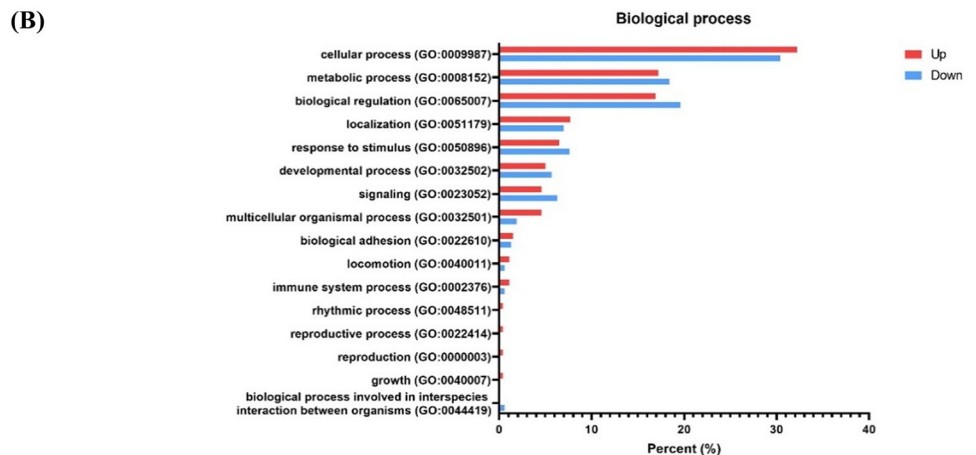

**(C)**

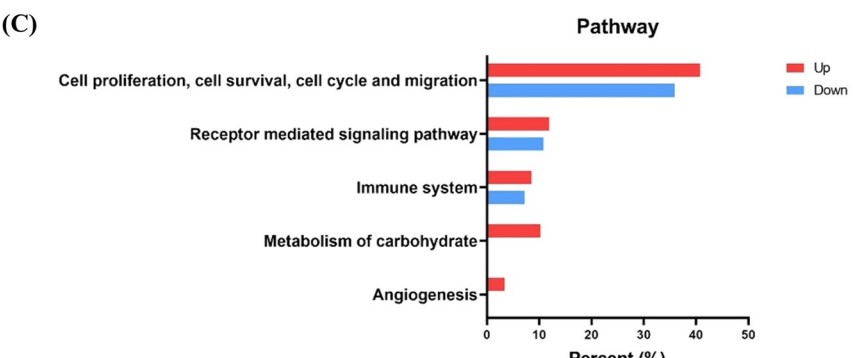

**Fig 8. PANTHER analysis performed a Gene Ontology (GO) annotation of the identified DEPs.** (A) Molecular function. (B) Biological process. (C) Pathway classification. The up-regulated DEPs are represented by red bars, while the down-regulated DEPs are represented by blue bars.

between the groups (Fig 11B). Additionally, the PCA (Fig 11C) revealed the grouping of each subtype in almost separate clusters. The samples in the luminal A subtype appeared closely grouped with the luminal B subtype, and the HER2 and TNBC samples were grouped close to

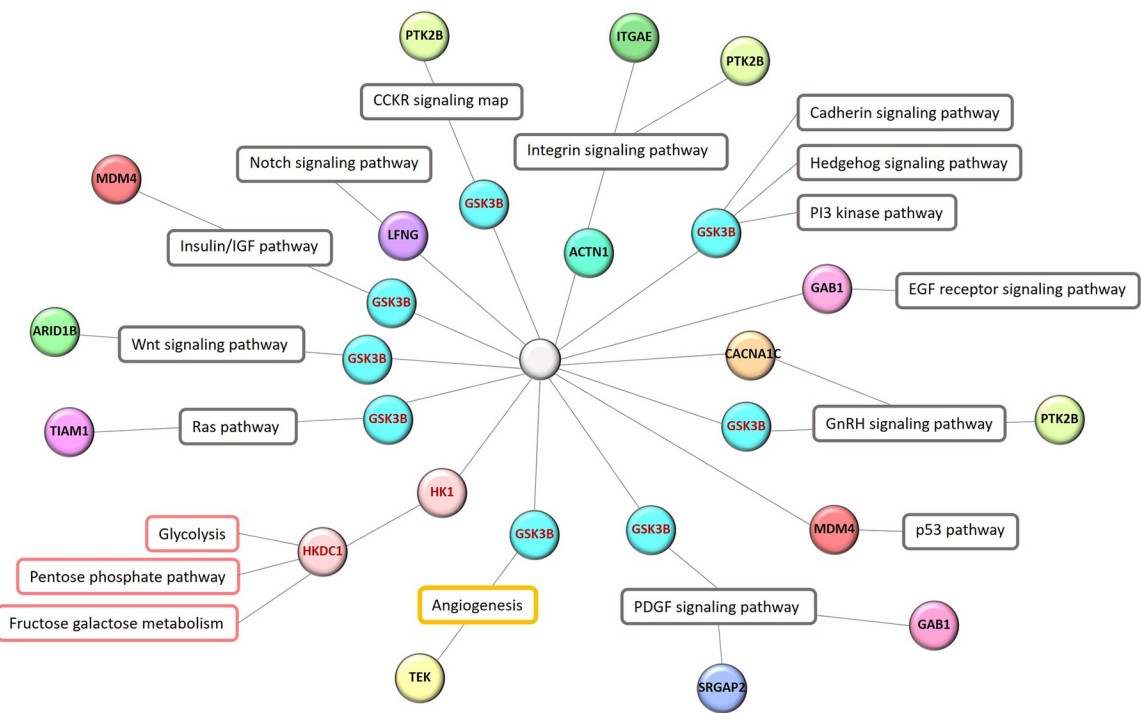

**Fig 9. Network representing the up-regulated DEPs in pathways associated with cancer progression, the metabolism of carbohydrates, and angiogenesis, as annotated by the PANTHER database.**

each other. We further analyzed the top-ranked hub proteins generated by the MCODE algorithm in each BC subtype. The four core protein groups were associated with the processes of RNA spicing, thermogenesis, the electron transport chain, and eukaryotic translation elongation, which were strongly expressed in the luminal A, luminal B, HER2, and TNBC subtypes, respectively (S3 Fig).

## Identification of stage-specific uEV proteins in BC

We compared the uEV protein profiles of a control group with those of three different BC stages (I–III) to investigate the distinct uEV protein signatures associated with each BC subtype. The evaluation revealed 558 unique proteins in the CT group, while samples from stage I, stage II, and stage III BC had 857, 1,202, and 74 unique proteins, respectively (Fig 12A and S5 Table). The heat map analysis unveiled distinctive patterns of protein expression across the groups. Nevertheless, the proteomic data of the BC samples showed overlapping profiles within each stage and with the control samples, confirming the PCA findings (Fig 12B and 12C). We analyzed unique proteins specific to each BC stage. The top one MCODE of stage I and stage II were proteins significantly involved with the ribosome and eukaryotic translation elongation, respectively. However, stage III BC did not exhibit any identifiable hub protein clusters (S3 Fig).

## Discussion

We demonstrated uEV concentration and proteomic profiling in 47 patients with BC and 29 healthy females in this study. One barrier to uEV isolation was the existing THP or uromodulin in urine as a contaminant. The filamentous network structure of THP entrapped uEVs, which decreased uEV yields. To eliminate THP contamination, DTT was applied to reduce

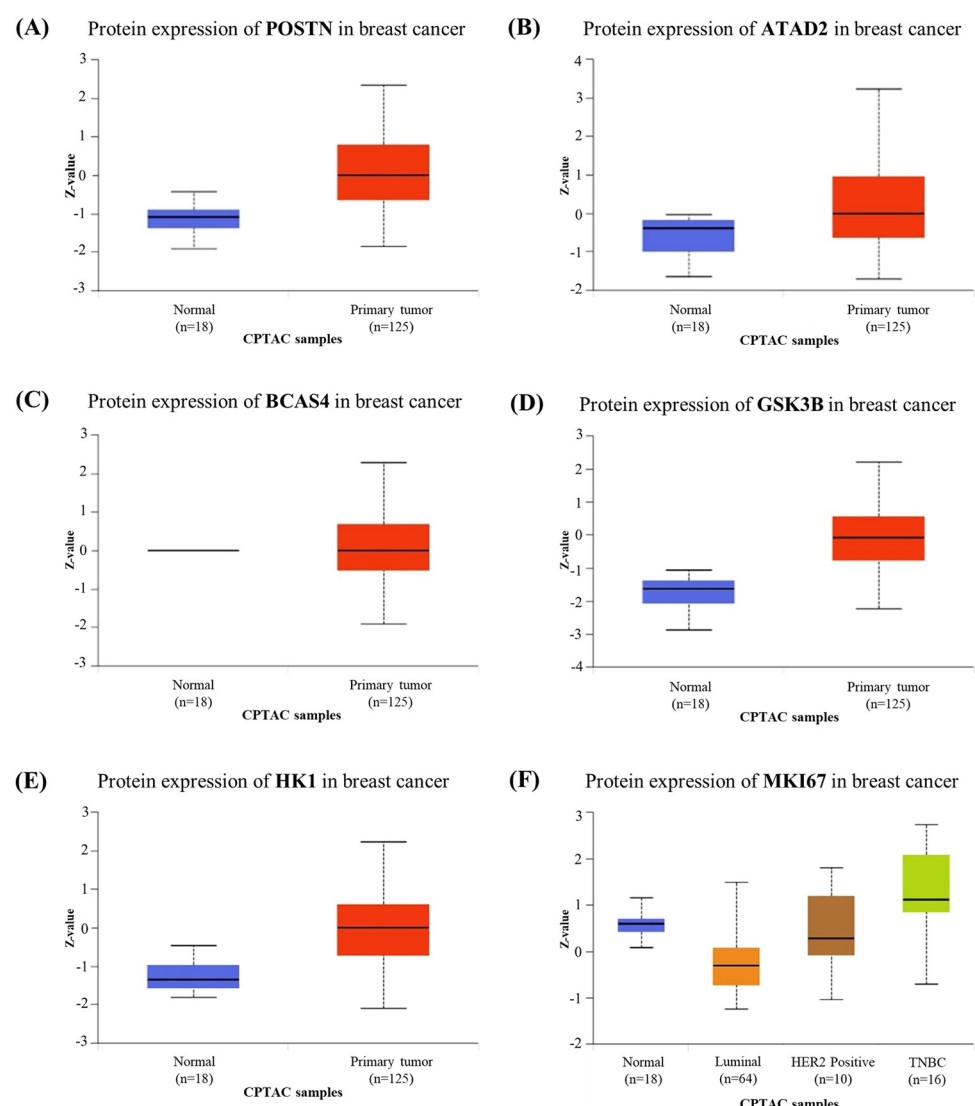

**Fig 10. Web-based validation of up-regulated DEPs.** Expression level of POSTN (A), ATAD2 (B), BCAS4 (C), GSK3B (D), and HK1 (E) in BC samples compared with normal samples. Expression level of Ki-67 (F) according to major BC subclass. Z-values represent standard deviations from the median across samples for the given cancer type. Log2 spectral count ratio values from the CPTAC were first normalized within each sample profile and then normalized across samples.

disulfide bond polymerization [25, 26], which significantly increased uEV yield [17]. The western blot analysis revealed that EV markers were identified in the uEV samples after the uEVs were isolated. Moreover, THP levels were found to be lower in the uEV samples when compared to those of the urine samples. This finding indicated that THP was removed by the incubation with DTT, supporting the experiments of Fernández-Llama et al. [27]. They recommended resuspending the pellet at a low speed and then treating it with 200 mg/ml DTT at 37˚C for 5–10 min to enrich the uEV markers. Besides, TEM has revealed that the uEVs were 30–150 nm in diameter and round-shaped, but the method was imprecise in terms of the typical cup shape of EVs. However, the NTA results confirmed that the mode size of the uEV samples was approximately 101–150 nm, and consistent with the size of small EVs. Taken together, these findings indicate that the obtained uEVs were well characterized and purified.

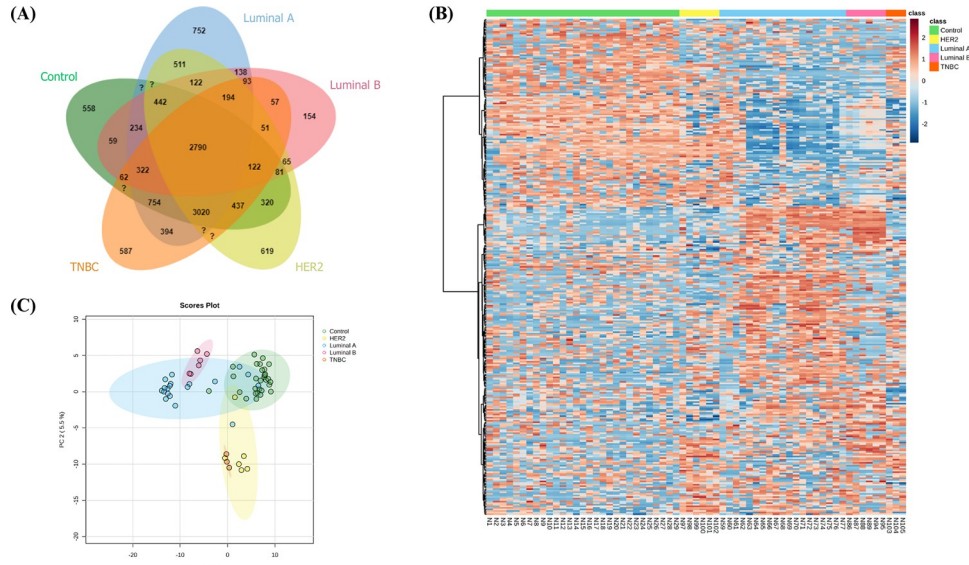

**Fig 11. uEV proteomic analysis among subtypes of BC patients compared with control samples.** (A) Venn diagram of all proteins identified in each sample group. (B) Heat map analysis of the uEV proteomic profile in individual samples. (C) PCA of the proteome of uEVs using 29 CT samples and 19, 6, 6, and 3 samples from BC patients with luminal A, luminal B, HER2-enriched, and TNBC subtypes, respectively.

To quantify the concentration of uEVs, we measured the concentration of the urine particles in whole urine compared with the uEV concentration following isolation. Although the urine particle concentration of the CT group was significantly higher than that of the BC patients, the uEV concentration and the mode size of the uEVs did not differ significantly

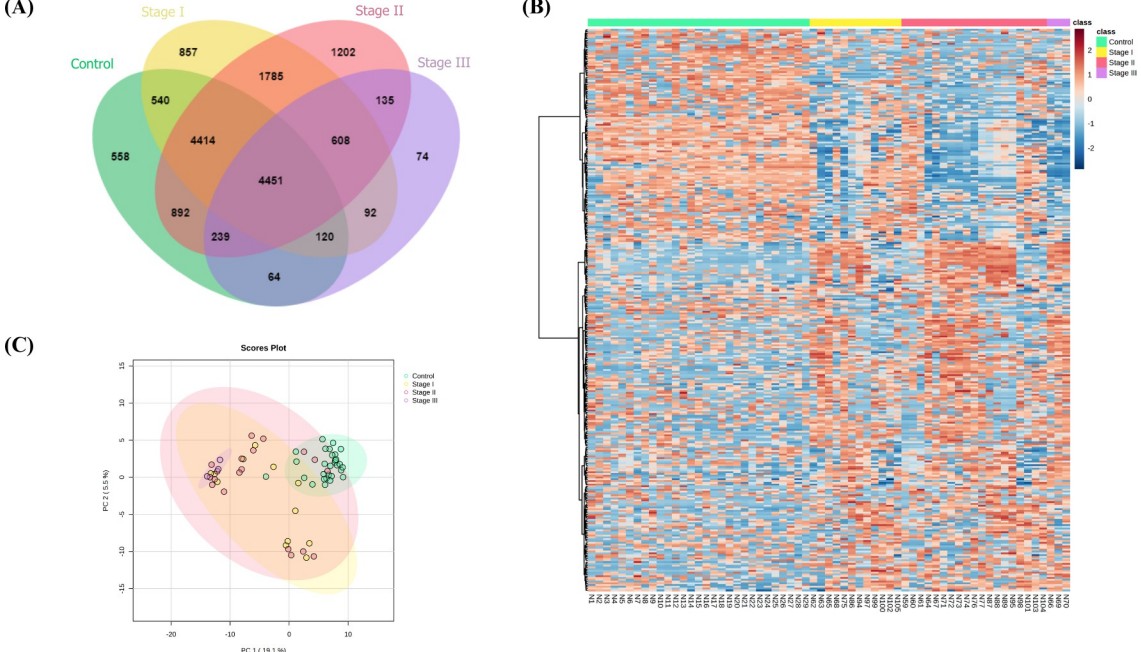

**Fig 12. uEV proteomic analysis among stage I–III BC patients compared with control samples.** (A) Venn diagram of all proteins identified in each sample group. (B) Heat map analysis of the uEV proteomic profile in individual samples. (C) PCA of the proteome of uEVs using 29 CT samples and 12, 19, and 3 samples from patients with stage I, II, and III BC, respectively.

between the two groups. It is possible that the CT urine contained other particles besides uEVs. Our results suggest that the concentration of urine particles and uEVs cannot sufficiently distinguish healthy women from BC patients. This is because urine concentration may be altered in both normal or pathological circumstances because urine volume is highly variable among individuals and depends on various factors, such as time of collection, food and water intake, medication, and exercise [28]. Therefore, these factors can affect urine particle concentration when measured by NTA. It is indeed possible that uEV concentrations may not differ between BC and healthy samples. Conversely, if cancer develops in the bladder, the uEV concentration between healthy donors and patients may substantially differ because the bladder is the primary cancer site. uEV concentrations have been reported as significantly higher in samples from patients with bladder cancer than in samples from healthy individuals [29]. However, uEV concentrations were also reported as higher in control patients in whom tumors were not observed [30]. The majority of studies on urinary EVs has focused on urological cancer and does not correlate with our findings. It was also reported that there was no significant difference in the uEV concentration between prostate cancer patients and healthy controls [31].

Strikingly, the protein profile of the uEVs was the highlight of this study. We discovered more than ten thousand of raw proteins in the uEVs, which were significantly greater than the total number of urinary proteins obtained from patients with BC and healthy females. Among the 155 up- and the 104 down-regulated proteins expressed in BC patients (when compared with those of the CT group). The PANTHER database was utilized to explain the function of the up- and down-regulated proteins. Remarkably, we discovered that the proteins related to cancer progression in the EVs derived from the urine of BC patients and the up-regulated DEPs were essential proteins for pathways implicated in cell proliferation, cell survival, the cell cycle, cell migration, carbohydrate metabolism, and angiogenesis (Fig 8C).

Multiple signaling pathways are driven by glycogen synthase kinase 3 beta (GSK-3β) (Fig 9). This protein is encoded by *GSK3B*, which regulates glycogen metabolism, represses Wnt signaling, and maintains cellular proliferation and apoptosis [32]. Depending on the tumor type, GSK-3β can be either a tumor promoter or suppressor [33]. Moreover, GSK-3β is important in neoplasia because it phosphorylates other key proteins mediating numerous pathways implicated in cancer (such as the Wnt/β-catenin, PI3K/PTEN/Akt/mTOR, Ras/Raf/MEK/ERK, Hedgehog, Notch, and TP53 pathways) that influence cancer initiation, epithelial–mesenchymal transition, and resistance to therapy [34]. Additionally, high levels of GSK-3β expression were associated with reduced relapse-free survival, while GSK-3β inhibitors were shown to suppress tumor growth in BC patients [35].

In our study, angiogenesis was indicated only by the up-regulated DEPs, in which GSK-3β can interact with angiopoietin-1 receptor or tyrosine kinase with Ig and EGF homology domains-2 (TIE2). TIE2 is encoded by *TEK* and acts as a cell-surface receptor for ANGPT1, ANGPT2, and ANGPT4, thereby regulating normal angiogenesis. TIE2 is overexpressed in the tumoral vessels of several cancer types, such as gastric tumors, breast tumors, and gliomas [36]. Moreover, Song *et al.* [37] showed that TIE2 expression and mRNA levels were higher in BC tissues than in normal tissues. The plasma levels of TIE2 were also reported as significantly higher in BC patients when compared with those of healthy subjects.

Energy is a necessary power source in cells undergoing rapid proliferation. Therefore, glycolysis is most certainly increased in several malignant tumors. HK is the first enzyme involved in the conversion of glucose to glucose-6-phosphate. It has four traditional isoforms (HK1, HK2, HK3, and HK4) encoded by different genes, as well as a newly discovered isoform, hexokinase domain-containing 1 (HKDC1) [38]. HK1 is associated with cancer progression, promotes cell proliferation, and is expressed in several cancer types. High expression levels of HK1 have been associated with a poor prognosis in patients with colorectal cancer [39],

ovarian cancer [40], and advanced-stage gastric cancer with lymphatic metastasis [41]. Moreover, HKDC1 expression was reported as significantly increased in BC cells and may potentially be involved in regulating breast tumorigenesis [42].

Web-based validation, as analyzed by UALCAN, was utilized to identify potential biomarkers, including six remarkable proteins that were up-regulated in the uEVs of the BC patients: periostin, ATAD2, BCAS4, GSK3β, HK1, and Ki-67. These proteins were strongly expressed in BC, mediated cancer progression and served as markers for cell proliferation.

Periostin is a matricellular protein encoded by the POSTN gene, that interacts with integrins and other extracellular matrix proteins such as collagens, fibronectin, tenascin C, and heparin, that are known to facilitate cancer development and progression through the promotion of cell proliferation, cell survival, angiogenesis, and the epithelial–mesenchymal transition (EMT). Periostin is overexpressed in many cancer types, including BC, lung cancer, colorectal cancer, and liver cancer. Elevated serum periostin is associated with poor prognosis in patients [43]. UALCAN notes that the expression of periostin in BC tumors is significantly higher than in normal tissues, while periostin can be overexpressed in stage I of BC (S2 Table). Our findings are supported by previous studies showing that periostin is enriched in exosomes deriving from the metastatic human BC cell line (MDA-MB-231) and the mouse BC cell line (4T1) as compared to those deriving from non-metastatic BC cell lines (MCF7 and 67NR). Moreover, periostin was mainly found to be expressed in the plasma exosomes of BC patients with lymph node (LN) metastasis as compared to BC patients with no LN metastasis [44]. The prognostic value of periostin was estimated in 259 tumors. In early stage BC patients receiving radiation therapy following conserving surgery, a multivariate analysis has revealed that periostin was linked to an elevated risk of local recurrence and distant metastasis [45]. Therefore, the expression of periostin in the uEVs of BC patients could be correlated to the aggressiveness and the metastasis of BC, and may be useful for early stage BC screening.

Liu *et al.* [46] have reviewed several studies that have reported that the ATAD2 can regulate different downstream molecules and is overexpressed in various cancer types, including BC, gastric cancer, pancreatic cancer, colorectal cancer, ovarian cancer, uterine corpus endometrial carcinoma, cervical cancer, prostate cancer, renal cancer, and lung cancer. Due to the fact that ATAD2 is a co-activator of the ERs and the androgen receptors in BC, it mediates the expression of E2 that induces cell proliferation and cell cycle progression in estrogen-dependent tumors [47]. Ciró *et al.* [48] have suggested that ATAD2 is associated with the E2F and MYC expression, thereby resulting in the promotion of an aggressive phenotype in cancer. As the oncogenic effects are closely related to pathogenesis, ATAD2 has emerged as a diagnostic and prognostic marker in numerous malignant tumors, and has attracted attention as a drug target for the inhibition of cancer progression. Therefore, the hereby generated evidence supports the hypothesis that ATAD2 could be a potential screening biomarker in BC.

Our findings have shown that in the uEVs isolated from BC patients, the proliferation marker Ki-67 was found to be significantly up-regulated (as compared to that of CT uEVs) with a high FC = 32.836. Ki-67 is a well-known marker of cell proliferation that is present in all proliferating cells during the G1, S, G2, and mitosis phases. It is associated with the cell proliferation activity used to indicate tumor aggressiveness in various cancer types [49]. Ki-67 is thought to be a crucial factor for the differentiation between luminal A and luminal B BC subtypes [50]. A high tumor grade and poor prognosis in BC patients are significantly related to a high Ki-67 index (≥15%) [51]. These studies have suggested that Ki-67 could reflect the promotion of cancer cell proliferation passed on uEVs that may be involved in BC pathogenesis.

Even though the fold change of BCAS4 was less than the other three up-regulated proteins (periostin, ATAD2, and Ki-67), it was still attractive because BCAS4 was significantly increased in breast tumors compared to normal tumors reported by UALCAN. As Bärlund

*et al.* described that breast carcinoma-amplified sequence 4 (*BCAS4*), as a gene at 20q13.2, which was commonly amplified in breast cancer and overexpressed in most breast cancer cell lines. However, the functions and transcription of BCAS4 are not clear, and the BCAS4 over-expression may be caused by the development of BC [52].

Through this comprehensive analysis, we expected to gain insights into the unique uEV protein signatures associated with each BC subtype and stage. However, we observed those of unique proteins associated with common pathways in cancer that might not be specific to each subtype and stage (S4 and S5 Figs). Furthermore, patients with the same disease stage had different subtypes, so the molecular subtypes and disease stages varied within individuals. Therefore, it was challenging to determine the main proteins expressed in a specific subtype and stage.

The ambiguity was revealed through the MCODE algorithm, which showed information regarding the hub proteins of TNBC. These hub proteins, including RPL10A, RPL39L, RPL27, RPS28, PSMC5, RPL38, RPS27L, and EEF1B2, belonged to the cluster of proteins expressed in the early stages (I–II) of BC (S3 Fig). Additionally, all of these proteins were associated with the ribosome and the elongation phase of protein synthesis. Ribosomes are essential for cell growth, proliferation, and development through protein translation [53]. Alteration of the expression of ribosomal proteins RPL10 and RPL39 has been linked to tumor initiation and BC progression. The 60S ribosomal protein L10A (RPL10A) is involved in cell growth, proliferation, and tumorigenesis, and it may play a critical role in BC progression [54]. In TNBC, RPL39 is important for stem cell self-renewal, therapy resistance, and lung metastases, and it promotes cancer through the inducible nitric oxide synthase signaling pathway [55, 56]. The ribosomal protein RPL27A is significantly upregulated in TNBC models, promoting TNBC development and metastasis through the eukaryotic initiation factor 2 signaling pathway. Targeting of RPL27A shows promise as a therapeutic strategy, reducing the extent of cell migration and invasion [57]. PSMB5, a crucial regulator of proteasome function, is related to poor prognoses in patients with BC [58]. It is associated with proliferation and drug resistance and is overexpressed in TNBC tissue. These findings suggest PSMB5 as a biomarker and therapeutic target for TNBC [59]. RPS28 has been identified as a potential prognostic gene candidate in early-stage hormone receptor-negative BC and TNBC [60]. RPS27L and RPL38 are RNA-binding proteins associated with BC, and their functions have been evaluated through machine-learning predictions. Additionally, RPL38 has been identified as a cancer immunotherapy protein and a potential target for immunotherapy in BC [61]. EEF1B2 was reported as downregulated in IR-induced senescence in MCF7 cells [62]. Conversely, increased EEF1B2 expression has been observed in the majority of cancer types [63].

Our analysis examined specific proteins in different BC subtypes and stages and provided valuable insights that also highlighted the uncertainty regarding protein specificity in BC subtypes and stages. These findings reflect the need for further investigation, and the number of samples in each group should be increased using a larger cohort. Taken together, the protein profiling of BC patient-derived uEVs exhibited characteristics of cancer development and cancer progression through several pathways (such as those involved in cell proliferation, cell survival, cell migration, cell cycle, glycolysis, and angiogenesis). Moreover, signature proteins were distinctively up-regulated in patients with BC thereby appearing as promising potential biomarkers for use in BC screening.

## Conclusions

The uEV concentration could not be used in order to distinguish BC patients from healthy women. Surprisingly, our study has identified thousands of uEV proteins that were up-

regulated and overexpressed in BC patients. Furthermore, some of these proteins belonged to pathways implicated in cancer progression, and their uEV expression was able to differentiate BC patients from healthy women. Our findings support the use of uEV proteins as screening biomarkers for BC.

## Supporting information

**S1 Fig. Concentration of total particles in urine and uEVs among BC subtypes and in samples from healthy controls.** (A) Total particle concentration in urine (median ± IQR). (B) Total particle size distribution in urine. (C) Mode size of total particles in urine according to BC subtype (median ± IQR). (D) uEV concentration (median ± IQR). (E) uEV size distribution. (F) Mode size of uEVs according to BC subtype (median ± IQR). Results were derived from 29 CT samples and 19, 6, 6, and 3 samples from BC patients with luminal A, luminal B, HER2-enriched, and TNBC subtypes, respectively. *: p < 0.05.
(TIF)

**S2 Fig. Concentration of total particles in urine and uEVs among BC stages and in samples from healthy controls.** (A) Total particle concentration in urine (median ± IQR). (B) Total particle size distribution in urine. (C) Mode size of total particles in urine according to BC subtype (median ± IQR). (D) uEV concentration (median ± IQR). (E) uEV size distribution. (F) Mode size of uEVs according to BC subtype (median ± IQR). Results were derived from 29 CT samples and 19, 6, 6, and 3 samples from BC patients with luminal A, luminal B, HER2-enriched, and TNBC subtypes, respectively. *: p < 0.05.
(TIF)

**S3 Fig. Top MCODE extracted from the unique uEV proteins for each particular subtype and stage of BC.** The hub proteins were selected from the protein-protein interaction network annotated by STRING operated by the MCODE algorithm using Cytoscape. The cutoff criteria were set as follows: degree cutoff = 2, node score cutoff = 0.2, k-core = 2, and maximum depth = 100.
(TIF)

**S4 Fig. PANTHER pathway of the unique uEV protein-specific subtypes of BC.** The top 15 pathways were determined using WEB-based GEne SeT AnaLysis Toolkit (WebGestalt).
(TIF)

**S5 Fig. PANTHER pathway of the unique uEV protein-specific stages of BC.** The top 15 pathways were determined using WEB-based GEne SeT AnaLysis Toolkit (WebGestalt).
(TIF)

**S1 Table. The 259 significantly DEPs (FC≥4) in uEVs of BC compared with CT.**
(DOCX)

**S2 Table. PANTHER pathway analysis of the up-regulated DEPs (FC≥4) in uEVs compared with CT.**
(DOCX)

**S3 Table. PANTHER pathway analysis of the down-regulated DEPs (FC≥4) in uEVs of samples from patients with BC compared with CT samples.**
(DOCX)

**S4 Table. UALCAN validation of potential biomarkers from significantly up-regulated uEV proteins in BC samples compared with CT samples.**
(DOCX)

**S5 Table. List of unique uEV proteins identified in patients with BC with different molecular subtypes and stages compared with CT samples.**
(DOCX)

**S1 Raw image.**
(PDF)

## Acknowledgments

We acknowledge Assoc. Prof. Dr. Hutcha Sriplung (Epidemiology Unit, Faculty of Medicine, Prince of Songkla University) for providing urine samples and Dr. Sittiruk Roytrakul (National Center for Genetic Engineering and Biotechnology; BIOTEC) for performing the LC-MS/MS analysis. This work was supported by the Department of Biomedical Sciences and Biomedical Engineering, Faculty of Medicine, Prince of Songkla University. We also thank RN lab members for their helpful assistance.

## Author Contributions

**Conceptualization:** Hutcha Sriplung, Sittiruk Roytrakul, Raphatphorn Navakanitworakul.

**Data curation:** Nilobon Jeanmard, Sittiruk Roytrakul.

**Formal analysis:** Nilobon Jeanmard, Rassanee Bissanum, Sittiruk Roytrakul, Raphatphorn Navakanitworakul.

**Funding acquisition:** Raphatphorn Navakanitworakul.

**Investigation:** Nilobon Jeanmard, Sittiruk Roytrakul, Raphatphorn Navakanitworakul.

**Methodology:** Nilobon Jeanmard, Rassanee Bissanum, Sawanya Charoenlappanit, Sittiruk Roytrakul, Raphatphorn Navakanitworakul.

**Project administration:** Raphatphorn Navakanitworakul.

**Software:** Sittiruk Roytrakul.

**Supervision:** Sittiruk Roytrakul, Raphatphorn Navakanitworakul.

**Visualization:** Raphatphorn Navakanitworakul.

**Writing – original draft:** Nilobon Jeanmard.

**Writing – review & editing:** Sittiruk Roytrakul, Raphatphorn Navakanitworakul.

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
