## [Decision Letter · Decision Letter 0]

22 May 2023

PONE-D-23-06083Proteomic profiling of urinary extracellular vesicles differentiates breast cancer patients from healthy womenPLOS ONE

Dear Dr. Navakanitworakul,

Thank you for submitting your manuscript to PLOS ONE. After careful consideration, we feel that it has merit but does not fully meet PLOS ONE’s publication criteria as it currently stands. Therefore, we invite you to submit a revised version of the manuscript that addresses the points raised during the review process.

We look forward to receiving your revised manuscript.

Kind regards,

Santosh Kumar Mishra

Academic Editor

PLOS ONE

2. You indicated that ethical approval was not necessary for your study. We understand that the framework for ethical oversight requirements for studies of this type may differ depending on the setting and we would appreciate some further clarification regarding your research. Could you please provide further details on why your study is exempt from the need for approval and confirmation from your institutional review board or research ethics committee (e.g., in the form of a letter or email correspondence) that ethics review was not necessary for this study? Please include a copy of the correspondence as an ""Other"" file.

“This research was supported by the National Science, Research and Innovation Fund (NSRF) and Prince of Songkla University (Grant No. SCI4693040) to RN.”

“No”

Reviewers' comments:

Reviewer's Responses to Questions

**Comments to the Author**

1. Is the manuscript technically sound, and do the data support the conclusions?

Reviewer #1: Yes

Reviewer #2: Partly

2. Has the statistical analysis been performed appropriately and rigorously? 

Reviewer #1: Yes

Reviewer #2: Yes

3. Have the authors made all data underlying the findings in their manuscript fully available?

Reviewer #1: Yes

Reviewer #2: Yes

4. Is the manuscript presented in an intelligible fashion and written in standard English?

Reviewer #1: Yes

Reviewer #2: No

5. Review Comments to the Author

Reviewer #1: This clinical research study characterized the urine EVs (uEV) in 47 breast cancer patients and 29 healthy females. uEV were characterized for purity and then for proteomics. Outcomes identified several protein biomarkers for breast cancer. The development of liquid biopsies for breast cancer is urgently needed, therefore results from this study are important. However, I have few issues with the study as outlined below:

1. Authors need to provide more information regarding urine sample collection: random or timed, container used etc

2. Authors should analyze the data based upon the cancer subtype- this could lead to identification of novel proteins in uEV specific to each cancer subtype.

3. Please present how many proteins in the proteomics data overlap with those reported in exosome/EV protein databases (ExoCarta, EVpedia etc)

4. Authors should validate a few of their top hits of proteomics data by ELISA or any secondary method.

Reviewer #2: In this paper, Nibolon Jeanmard et al. report on a proteomic analysis of urinary Extracellular Vesicles (uEVs) isolated from 29 healthy women and 47 women with breast cancer. They compare the concentration and proteome composition of uEVs isolated by ultracentrifugation following removal of uromodulin by DTT treatment, as well as of total particles in the mid-flow urine of women in each group. They identify a set of 155 proteins overrepresented in breast cancer patients’ urine and 104 proteins underrepresented. Finally, the authors performed a functional analysis of the identified proteins and discuss their relevance to cancer in general and breast cancer in particular.

Comments:

1) The text needs thorough editing; it is speckled with grammatical mistakes and typos.

2) The rational for comparing EVs in total urine (cleared of cells and large cellular debris by low-speed centrifugation is not clear and poorly justified.

3) Figure Legends are minimal; they need to describe the figures better.

4) The author used only DTT to rid of THP (Uromodulin). While this is a recognized and published method when coupled to ultracentrifugation, it is not the most efficient method for THP removal. This is clearly shown by the presence of THP in uEVs lines 1 and 2. The publication of DOI:10.1002/prca.201900018

5) This reviewer regrets the lack of a more profound analysis of the proteomic data in relation to breast cancer stages or tumor size. Such analysis may have been saved for a subsequent submission. The authors have the unique opportunity to test their hypothesis by comparing the uEVs in breast cancer early stages (I), small tumor sizes (<2cm), and lack of lymph node involvement with that of healthy controls. This would strengthen their assumption that uEVs can be used for non-invasive early breast cancer detection.

6) While the authors perform functional analysis of the proteins they found modulated in breast cancer patients, they do not compare their data with data from other urine studies in Vesiclepedia.

6. PLOS authors have the option to publish the peer review history of their article (what does this mean?). If published, this will include your full peer review and any attached files.

Reviewer #1: No

Reviewer #2: **Yes: **Jean-Charles Grivel

---

## [Author Response · Author response to Decision Letter 0]

12 Jul 2023

Reviewer #1: This clinical research study characterized the urine EVs (uEV) in 47 breast cancer patients and 29 healthy females. uEV were characterized for purity and then for proteomics. Outcomes identified several protein biomarkers for breast cancer. The development of liquid biopsies for breast cancer is urgently needed, therefore results from this study are important. However, I have few issues with the study as outlined below: 

Response to Reviewer#1: Thank you for your valuable suggestions on how to enhance the impact of our findings in the field of cancer research. Below is a point-by-point response to your comments and concerns. Line numbers as they appear in the revised manuscript are added for your reference. We have also highlighted the changes with red letters within the revised manuscript.

1. Authors need to provide more information regarding urine sample collection: random or timed, container used etc 

Response to Reviewer#1: Thank you for your comment. We have added more information to lines 95–96.

2. Authors should analyze the data based upon the cancer subtype- this could lead to identification of novel proteins in uEV specific to each cancer subtype. 

Response to Reviewer#1: Thank you for your insightful suggestion. The concentration of urinary extracellular vesicles (uEVs) did not significantly differ among the breast cancer (BC) subtypes or when compared with healthy control (CT) samples (see S1 Fig). Proteomic analysis revealed that the unique proteins of each subtype were common proteins in cancer development pathways. In particular, these proteins were involved in ribosome and protein synthesis, and some were expressed in specific subtypes. However, when analyzed according to the stage, as suggested by Reviewer 2, we found that these proteins might also be expressed in the early stages of BC. Therefore, it may be challenging to define the stage and subtype of BC based on uEV proteomics. Thus, there is an urgent need for validation of our study findings through machine learning or with a larger cohort. We have added this information to the Results and Discussion sections (lines 231–232, 363–375, and 522–561).

3. Please present how many proteins in the proteomics data overlap with those reported in exosome/EV protein databases (ExoCarta, EVpedia etc)

Response to Reviewer#1: As suggested, the number of uEV proteins in the proteomics data that overlapped with uEV proteins in the EV protein databases (ExoCarta, EVpedia, and Vesiclepedia) is shown in Fig 4 and explained in the Results section of the revised manuscript (lines 251–256).

4. Authors should validate a few of their top hits of proteomics data by ELISA or any secondary method. 

Response to Reviewer#1: Thank you for your valuable advice. Although promising uEV protein biomarkers were identified in our discovery phase, we were unable to validate their presence in clinical samples because the number of samples was insufficient. Additionally, there was a lack of specific antibodies to perform western blot using cell lines to validate our findings in case subtypes because these antibodies require at least 3 months for manufacture.

Reviewer #2: In this paper, Nibolon Jeanmard et al. report on a proteomic analysis of urinary Extracellular Vesicles (uEVs) isolated from 29 healthy women and 47 women with breast cancer. They compare the concentration and proteome composition of uEVs isolated by ultracentrifugation following removal of uromodulin by DTT treatment, as well as of total particles in the mid-flow urine of women in each group. They identify a set of 155 proteins overrepresented in breast cancer patients’ urine and 104 proteins underrepresented. Finally, the authors performed a functional analysis of the identified proteins and discuss their relevance to cancer in general and breast cancer in particular.

Response to Reviewer#2: Thank you for your valuable suggestions on how to enhance the impact of our findings in the field of cancer research. Below is a point-by-point response to your comments and concerns. Line numbers as they appear in the revised manuscript are added for your reference. We have also highlighted the changes with red letters within the revised manuscript.

1) The text needs thorough editing; it is speckled with grammatical mistakes and typos.

Response to Reviewer#2: The manuscript has been carefully reviewed by Enago for grammatical errors and spelling mistakes.

2) The rational for comparing EVs in total urine (cleared of cells and large cellular debris by low-speed centrifugation is not clear and poorly justified.

Response to Reviewer#2: As shown in the literature review, uEV concentration in the cancer group was greater than in the CT group. Thus, we hypothesized that uEV concentrations in BC samples would be significantly higher than those in the CT group. And is there the possibility of assessing uEV measurement in whole urine samples (cleared of cells) without UC? Therefore, we demonstrated this in both types of samples.

3) Figure Legends are minimal; they need to describe the figures better.

Response to Reviewer#2: We have added more details to the figure legends.

4) The author used only DTT to rid of THP (Uromodulin). While this is a recognized and published method when coupled to ultracentrifugation, it is not the most efficient method for THP removal. This is clearly shown by the presence of THP in uEVs lines 1 and 2. The publication of DOI:10.1002/prca.201900018 

Response to Reviewer#2: Thank you for your helpful comment. We agree that ultracentrifugation (UC) may not be the most efficient method for THP removal. We attempted to remove as much THP as possible using a combination of isolation solution (10 mM triethanolamine and 250 mM sucrose) and DTT (see lines 125–129). Unfortunately, THP was not completely eliminated from the uEV samples and was only partially decreased. Even though uEV isolation via UC is time-consuming, it is widely used in uEV proteome discovery (see Table 3 of Merchant et al., 2017, https://doi.org/10.1038/nrneph.2017.148). Our laboratory already has a UC set-up in place, and it is the main protocol for EV isolation. We expect to use this urinary EV isolation protocol in the large-scale clinical study, and it is a simple and inexpensive technique. Following your recommendation, we will consider the use of hydrostatic filtration dialysis in further studies.

5) This reviewer regrets the lack of a more profound analysis of the proteomic data in relation to breast cancer stages or tumor size. Such analysis may have been saved for a subsequent submission. The authors have the unique opportunity to test their hypothesis by comparing the uEVs in breast cancer early stages (I), small tumor sizes (<2cm), and lack of lymph node involvement with that of healthy controls. This would strengthen their assumption that uEVs can be used for non-invasive early breast cancer detection.

Response to Reviewer#2: Thank you for your thoughtful advice. The uEV concentrations did not differ between each BC stage or when compared with CT samples (see S2 Fig).

We used the Molecular Complex Detection plugin within Cytoscape v3.10.0 to analyze the top-ranked unique hub proteins in patients with early-stage (I–II) BC. Then, we selected the top one MCODE from the unique uEV proteins particular to each subtype and stage of BC, which included various ribosomal proteins and proteins involved in translation. Our findings may aid the development of non-invasive methods for early detection of BC. We have revised the Results and Discussion sections accordingly (lines lines 231–232, 363–375, and 522–561).

6) While the authors perform functional analysis of the proteins they found modulated in breast cancer patients, they do not compare their data with data from other urine studies in Vesiclepedia. 

Response to Reviewer#2: The number of uEV proteins in the proteomics data that overlapped with uEV proteins in the EV protein databases (ExoCarta, EVpedia, and Vesiclepedia) is shown in Fig. 4 and explained in the Results section of the revised manuscript (lines 251–256).

---

## [Decision Letter · Decision Letter 1]

1 Sep 2023

Proteomic profiling of urinary extracellular vesicles differentiates breast cancer patients from healthy women

PONE-D-23-06083R1

Dear Dr. Navakanitworakul,

We’re pleased to inform you that your manuscript has been judged scientifically suitable for publication and will be formally accepted for publication once it meets all outstanding technical requirements.

Kind regards,

Santosh Kumar Mishra

Academic Editor

PLOS ONE

Additional Editor Comments (optional):

Reviewers' comments:

Reviewer's Responses to Questions

**Comments to the Author**

1. If the authors have adequately addressed your comments raised in a previous round of review and you feel that this manuscript is now acceptable for publication, you may indicate that here to bypass the “Comments to the Author” section, enter your conflict of interest statement in the “Confidential to Editor” section, and submit your "Accept" recommendation.

Reviewer #1: All comments have been addressed

Reviewer #2: All comments have been addressed

2. Is the manuscript technically sound, and do the data support the conclusions?

Reviewer #1: Yes

Reviewer #2: Yes

3. Has the statistical analysis been performed appropriately and rigorously? 

Reviewer #1: Yes

Reviewer #2: Yes

4. Have the authors made all data underlying the findings in their manuscript fully available?

Reviewer #1: Yes

Reviewer #2: Yes

5. Is the manuscript presented in an intelligible fashion and written in standard English?

Reviewer #1: Yes

Reviewer #2: Yes

6. Review Comments to the Author

Reviewer #1: (No Response)

Reviewer #2: The authors have properly answered all the concerns raised by the reviewer. They have performed the requested additional analysis and provided adequate figures to illustrate this analysis.

They have discussed the new analysis.

7. PLOS authors have the option to publish the peer review history of their article (what does this mean?). If published, this will include your full peer review and any attached files.

Reviewer #1: No

Reviewer #2: **Yes: **Jean-Charles Grivel

---

## [Editor Report · Acceptance letter]

27 Oct 2023

PONE-D-23-06083R1 

Proteomic profiling of urinary extracellular vesicles differentiates breast cancer patients from healthy women 

Dear Dr. Navakanitworakul:

I'm pleased to inform you that your manuscript has been deemed suitable for publication in PLOS ONE. Congratulations! Your manuscript is now with our production department. 

Kind regards, 

on behalf of

Dr. Santosh Kumar Mishra 

Academic Editor

PLOS ONE